# l-Lactic Acid Production Using Engineered *Saccharomyces cerevisiae* with Improved Organic Acid Tolerance

**DOI:** 10.3390/jof7110928

**Published:** 2021-10-31

**Authors:** Byeong-Kwan Jang, Yebin Ju, Deokyeol Jeong, Sung-Keun Jung, Chang-Kil Kim, Yong-Suk Chung, Soo-Rin Kim

**Affiliations:** 1Major in Food Application Technology, School of Food Science and Biotechnology, Kyungpook National University, Daegu 41566, Korea; byeonggwan_jang@naver.com (B.-K.J.); jyb52795279@gmail.com (Y.J.); dyj@knu.ac.kr (D.J.); skjung04@knu.ac.kr (S.-K.J.); 2Department of Horticulture, Kyungpook National University, Daegu 41566, Korea; ckkim@knu.ac.kr; 3Department of Plant Resources and Environment, Jeju National University, Jeju 63243, Korea

**Keywords:** lactate dehydrogenase, whole-genome sequencing, *Saccharomyces cerevisiae*, lignocellulosic biomass, polylactic acid

## Abstract

Lactic acid is mainly used to produce bio-based, bio-degradable polylactic acid. For industrial production of lactic acid, engineered *Saccharomyces cerevisiae* can be used. To avoid cellular toxicity caused by lactic acid accumulation, pH-neutralizing agents are used, leading to increased production costs. In this study, lactic acid-producing *S. cerevisiae* BK01 was developed with improved lactic acid tolerance through adaptive laboratory evolution (ALE) on 8% lactic acid. The genetic basis of BK01 could not be determined, suggesting complex mechanisms associated with lactic acid tolerance. However, BK01 had distinctive metabolomic traits clearly separated from the parental strain, and lactic acid production was improved by 17% (from 102 g/L to 119 g/L). To the best of our knowledge, this is the highest lactic acid titer produced by engineered *S. cerevisiae* without the use of pH neutralizers. Moreover, cellulosic lactic acid production by BK01 was demonstrated using acetate-rich buckwheat husk hydrolysates. Particularly, BK01 revealed improved tolerance against acetic acid of the hydrolysates, a major fermentation inhibitor of lignocellulosic biomass. In short, ALE with a high concentration of lactic acid improved lactic acid production as well as acetic acid tolerance of BK01, suggesting a potential for economically viable cellulosic lactic acid production.

## 1. Introduction

Lactic acid is an organic acid widely used in the food, pharmaceutical, textile, and chemical industries, especially for the production of polylactic acid [1,2], a biodegradable and compostable bioplastic [3,4,5]. Lactic acid is produced by microbial fermentation such as natural lactic acid producer, lactic acid bacteria (LAB) [6,7]. However, lactic acid production by natural producers is limited due to low maximum titers and optical isomers [8]. Additionally, LAB is not considered as a suitable host for industrial fermentations due to phage contamination issues and high nutritional requirements [9,10].

Engineered *Saccharomyces cerevisiae* expressing a heterologous lactate dehydrogenase gene (*ldh*) is a promising workhorse for industrial lactic acid production [11,12]. Various metabolic engineering approaches have been performed to improve lactic acid production by the yeast [13]. Expression of multiple copies of *ldh* genes increased lactate dehydrogenase expression levels and lactic acid production [14,15,16,17,18]. Additionally, rational engineering strategies have been used to convert more pyruvate to lactic acid by removing genes in the ethanol pathway, including pyruvate decarboxylase genes (*PDC1*, *PDC2*, *PDC5*, and *PDC6*) and alcohol dehydrogenase genes (*ADH1*, *ADH5*, and *ADH6*), which compete with the newly introduced lactic acid pathway [15,17,18,19,20,21]. 

Lactic acid causes cellular toxicity at near its pKa value (3.86) [22]. The use of neutralizing agents such as calcium hydroxide (Ca(OH)_2_) and calcium carbonate (CaCO_3_) during lactic acid fermentation therefore helps to avoid the toxicity and the accumulation of a high concentration of lactic acid [23]. However, when the fermentation is completed, the fermentation medium should be acidified for purification of lactic acid, which results in gypsum (CaSO_4_) formation and increased production costs [24,25]. Therefore, development of a lactic acid-tolerant strain is one of the most important strain engineering goals for industrial lactic acid production [12,17,18,26].

In this study, we performed adaptive laboratory evolution (ALE) of lactic acid-producing *S. cerevisiae* strain in a high concentration of lactic acid to develop an evolved mutant with improved tolerance against lactic acid. Using genome sequencing and metabolomic profiling, the evolved mutant was characterized. Using buckwheat husk hydrolysates, we demonstrated cellulosic lactic acid production by the evolved strain.

## 2. Materials and Methods

### 2.1. Strain Construction

Lactic acid-producing *S. cerevisiae* SR8LDH strain (Table 1) was developed in a previous study [8,27,28]. Briefly, the *ldh* gene from *Lactobacillus acidophilus* ATCC4356 expressed under the control of the *PGK1* promoter and the expression cassette were genome-integrated into the xylose-fermenting *S. cerevisiae* SR8 strain [29].

### 2.2. ALE

ALE was conducted in 100 mL Erlenmeyer flasks with 20 mL complex medium (10 g/L yeast extract and 20 g/L peptone) containing 20 g/L glucose and 8% (*w/v*) lactic acid (pH 3.1). The initial cell density was 0.05 g dry cell weight (DCW)/L. When cells entered the stationary phase, 1% culture was transferred to new media. Serial subcultures were performed until there was no change in growth rate. From the final culture, six colonies were isolated and evaluated for growth rate in complex medium containing 20 g/L glucose and 8% lactic acid.

### 2.3. Fermentation Studies

A single colony was pre-cultivated in 5 mL complex medium containing 20 g/L glucose at 30 °C and 250 rpm for 24 h. Fermentations were performed in 100 mL Erlenmeyer flasks with 20 mL complex medium containing 20 g/L or 200 g/L glucose (pH 5.7) under 30 °C, oxygen-limited conditions (80 rpm). All fermentations were independently performed in triplicate without pH control.

### 2.4. Genome Sequencing and Single-Nucleotide Polymorphism (SNP) Analysis

Genomic DNA from the cells was extracted using YeaStar DNA kits (Zymo Research, Irvine, CA, USA) and purified using Genomic DNA purification kits (Promega, Madison, WI, USA) for genome sequencing. DNA quantity and quality were confirmed using the Quant-It PicoGreen dsDNA assay kit (Invitrogen, Waltham, MA, USA) and agarose gel electrophoresis, respectively. Barcoded library construction and genome sequencing using Illumina Miseq instrumentation (Illumina, San Diego, CA, USA) were performed at CHUNLAB (CHUNLAB, Seoul, Korea). Barcoded shotgun genomic DNA libraries were constructed using the TruSeq Sample Prep Kit, following the manufacturer’s instructions (Illumina), and were subjected to 300 base pair (bp) paired-end sequencing by Illumina Miseq. Sequencing raw data were deposited in the Sequence Read Archive and are available at the NCBI BioProject PRJNA690663. SNP analyses were performed using CLC Genomics Workbench version 5.1 (QIAGEN, Hilden, Germany). Reads were trimmed based on quality scores from default program settings. The reads were mapped to an S288C yeast reference sequence (obtained from Genbank) to identify SNPs in SR8LDH and BK01. Next, unique, non-synonymous SNPs in BK01 relative to SR8LDH were identified (Appendix A).

### 2.5. SNP Confirmation by Sanger Sequencing

The genes with potential SNPs were amplified using Q5® High-Fidelity DNA Polymerase (New England Biolabs, Ipswich, MA, USA) according to manufacturer’s instructions. We used 20 ng/μL genomic DNA from SR8LDH and BK01 as template DNA. Amplified products were purified using Geneall ExpinTM PCR SV kit (Geneall, Seoul, Korea). Sanger sequencing was performed to confirm predicted SNPs (Cosmogenetech, Seoul, Korea).

### 2.6. Reverse Engineering

Reverse engineering was performed by introducing two confirmed SNPs, m*YPT7* (533G>A) and m*YOL159C-A* (172C>A), into the parental strain (SR8LDH), as described previously using CRISPR-Cas9 [30]. Additionally, *YPT7*, *YOL159C-A*, and both genes were deleted equally from SR8LDH and BK01, respectively. Briefly, strains expressing the pRS41N-Cas9 plasmid [31] were transformed with the guide RNA plasmids (Appendix A) and the donor DNA fragments prepared by using primers listed in Appendix A. Exceptionally, the m*YOL159C-A* allele could not be introduced directly using CRISPR-Cas9. Therefore, the wild copy of *YOL159C-A* was deleted, and the resulting knockout mutant was transformed to introduce the m*YOL159C-A* allele, amplified from the genomic DNA of BK01. Using the primers in Appendix A and sanger sequencing, the reverse-engineered strains were confirmed as the SR8LDH_m*YPT7* strain (LDH_R1), SR8LDH_m*YOL159C-A* strain (LDH_R2) strain, and the SR8LDH_m*YPT7*/m*YOL159C-A* strain (LDH_R3) (Table 1). 

### 2.7. Cell growth, pH, and Extracellular Metabolite Analysis

Cell density was presented by DCW, which was converted from the absorbance at 600 nm using a spectrophotometer (OPTIZEN NANO Q, Mecasis, Daejeon, Korea) (OD_600_ 1 = 0.47 g DCW/L). A pH meter (SevenCompact pH meter S220, Mettler Toledo, Columbus, OH, USA) was used for the pH determination of hydrogen ion concentration. Glucose, l-lactic acid, and ethanol concentrations during fermentation were analyzed, as previously described [32]. Briefly, serially diluted cultures were centrifuged at 13,000 rpm for 10 min at 4 °C. Supernatants were analyzed by high performance liquid chromatography (1260 Series, Agilent Technologies, CA, Santa Clara, USA) equipped with a refractive detector, connected to a Rezex ROA-Organic Acid H+ (8%) column (Phenomenex Inc., Torrance, CA, USA). The column was eluted in 0.005 N H_2_SO_4_ at a flow rate of 0.6 mL/min at 50 °C. The retention time for glucose, ethanol, and lactic acid was 4.938 min, 10.873 min, and 6.960 min, respectively.

### 2.8. Intracellular Metabolite Extraction and Derivatization

Cell quenching, intracellular metabolite extraction, and derivatization for intracellular metabolite analyses were performed as previously described [33,34]. Briefly, 5 mL of cell culture at the mid-exponential growth phase was quickly injected and quenched in 25 mL 60% (*v/v*) cold methanol with 10 mM HEPES (pH 7.1) at −40 °C. Quenched cells were centrifuged, and supernatants were discarded. Then, 1 mL of 75% (*v/v*) boiling ethanol with 10 mM HEPES (pH 7.1) was added to the quenched cell pellet and resuspended. The mixture was incubated at −80 °C for 5 min, then centrifuged, and extracted intracellular metabolites (supernatant) were collected. Extracts were vacuum-dried for 4 h using a speed vacuum concentrator (VC-96R, TAITEC, Saitama, Japan). Dried intracellular extracts underwent methoxyamination with 10 μL methoxyamine hydrochloride in pyridine (40 mg/mL, Sigma-Aldrich, St. Louis, MO, USA) and incubated at 30 °C for 90 min. For silylation, 40 μL N-Methyl-N-trimethylsilyl-trifluoroacetamide (Sigma-Aldrich) was added to samples and incubated at 37 °C for 30 min.

### 2.9. Intracellular Metabolite Analysis and Identification Using Gas Chromatography-Mass Spectroscopy (GC-MS)

GC-MS analysis was performed as previously described with some modifications [35]. An Agilent 6890 GC (Agilent Technologies) coupled to an Agilent 5973N MSD (Agilent Technologies) was used. One microliter of the derivatized sample was injected through an Agilent 7683 ALS (Agilent Technologies) into the GC in splitless mode. Samples were separated on an RTX-5Sil MS column (30 m × 0.25 mm × 0.25 μm, Restek, Bellefonte, PA, USA). The initial oven temperature was 50 °C for 1 min and then ramped at 20 °C/min to a final temperature of 330 °C and held for 5 min. Helium was used as a carrier gas at 0.7 mL/min. The temperatures of the ion source and transfer line were set to 230 °C and 250 °C, respectively. An electron impact of 70 eV was used for ionization. The mass selective detector was operated in scan mode with a mass range of 50–650 m/z.

### 2.10. Data and Statistical Analysis

To identify metabolites, raw data files were converted to netCDF format using Agilent Chemstation software (Agilent Technologies and the analysis base file converter; https://www.reifycs.com/AbfConverter/). Converted ABF format files were analyzed using MS-DIAL software (version: 4.60, Riken, Kanagawa, Japan) [36,37]. An average peak width of 20 scans and a minimum peak height of 1000 amplitudes were used for peak detection. A Sigma window value of 0.5 and electron ionization (EI) spectra cut-off of 10 amplitudes were used for deconvolution. For identification settings, the retention index tolerance was 2000, the m/z tolerance was 0.5 Da, the EI similarity cutoff was 70%, and the identification score cutoff was 80%. In the alignment parameter setting process, the retention time factor was 0.5. The peak intensity of identified metabolites was normalized by cell concentration (g DCW/L) for each sample. Statistica software (version 7.1, StatSoft, NYC, NY, USA) was used for principal component analysis (PCA), which is a multivariate analysis method and a useful statistical technique for reducing the dimensions of a dataset containing a large number of variables without the prerequisite knowledge of the dataset, by transforming original variables into abbreviated principal components (PCs) [38].

### 2.11. Lignocellulosic Hydrolysate Preparation and Simultaneous Saccharification and Fermentation (SSF) of the Engineered Yeast Strain

Buckwheat (*Fagopyrum esculentum* var*. Daesan*) was cultivated in Jeju Island (Korea) and harvested at the end of June in 2020. Buckwheat husks were separated from grains and dried at 60 °C for 24 h. The dried buckwheat husk samples were milled and stored at −80 °C until use. Buckwheat husk hydrolysate was prepared as previously described [39]. Four grams of buckwheat husk powder was mixed with 1% (*w/v*) H_2_SO_4_ solution, treated at 121 °C for 30 min by autoclave (MLS-3781L, Panasonic, Osaka, Japan), and neutralized using 7.5 N NaOH solution until pH 6.5. SSF of the buckwheat husk hydrolysates was performed as previously described [8,39]. For cellulase, Cellic® CTec2 (Novozymes, Bagsværd, Denmark) was used. A total of 40 filter paper cellulase units (FPU/g biomass) and 10 g DCW/L pre-cultured cells were added to the pretreated buckwheat husk hydrolysates, and the final volume was adjusted to 20 mL for 20% (*w/v*) solid SSF. Each fermentation was performed in 100 mL Erlenmeyer flasks for 24 h at 30 °C at 130 rpm under oxygen-limited conditions in biological triplicates.

## 3. Results and Discussion

### 3.1. ALE of Lactic Acid-Producing S. cerevisiae in a High Concentration of Lactic Acid

To improve lactic acid tolerance of the lactic acid-producing *S. cerevisiae* SR8LDH strain, cells were subjected to ALE by serial sub-cultures in complex medium containing a high concentration of lactic acid (8%), which critically limits cell growth [40,41]. During the first culture, a significantly long lag phase (216 h) was observed (Figure 1A). However, the lag time was greatly reduced in the second culture, with no more reduction in subsequent cultures. We isolated six colonies from the final culture and evaluated their lactic acid tolerance (Figure 1B). The most representative colony (BK01) was selected, and the improved lactic acid tolerance was confirmed again by the spotting assay (Figure 1C). SR8LDH and BK01 showed no difference in growth on complex agar medium containing 20 g/L glucose and 6% lactic acid. However, when lactic acid concentration was increased to 8%, BK01 showed better growth than SR8LDH.

In previous studies, ALE was also successfully applied to develop lactic acid-tolerant *S. cerevisiae* [17,18,42]. For the evolution conditions, lactic acid concentration gradually increased from 1% to 4% during 11 subcultures, and the evolved mutant showed no growth reduction with 1% lactic acid [17,18]. In another study, evolution was performed under 8% lactic acid conditions, and the resulting strain showed a maximum tolerance to 6% lactic acid and not growth on 8% lactic acid [42]. In the present study, BK01 was able to grow on 8% lactic acid, which was a higher tolerance than the previously reported strains. 

### 3.2. Genome Sequencing of the Evolved Strain and Reverse Engineering

To determine the genetic basis of the improved lactic acid tolerance, genome re-sequencing of the parental (SR8LDH) and the evolved (BK01) strains was performed, and 24 SNPs were identified, as listed in Appendix A. Sanger sequencing of the potential SNPs was able to confirm two SNPs in the m*YPT7* (533G>A) and m*YOL159C-A* (172C>A) genes. For reverse engineering, the two SNPs were introduced to SR8LDH, resulting in LDH_R1 (m*YPT7*), LDH_R2 (m*YOL159C-A*), and LDH_R3 (m*YPT7* and m*YOL159C-A*) (Table 1). However, those reverse-engineered strains did not show tolerance to lactic acid (Figure 1D). Based on the assumption of the loss of function mutations on the two genes (*YPT7*, and *YOL159C-A)* in BK01, those genes were deleted in SR8LDH, resulting in LDH_D1, LDH_D2, and LDH_D3 strains (Table 1), and those deletion mutants did not show tolerance to lactic acid (Appendix A). However, when any of the identified mutant alleles (m*YPT7* and m*YOL159C-A*) were deleted in BK01, the resulting strains (BK_D1, BK_D2, and BK_D3 in Table 1) lost the lactic acid tolerance. These results suggested that both m*YPT7* and m*YOL159C-A* are associated with the improved lactic acid tolerance of BK01 and that there are other genetic factors required for the tolerance, which could not be identified by simple SNP identification.

Among previously reported evolved *S. cerevisiae* strains on lactic acid, one strain was genome-sequenced, and the SNPs were identified in five different genes (*BSD2*, *ERF2*, *CIT2*, *NCL1*, and *SUR1*) [17]. Reverse engineering of the individual and combinatorial mutations discovered that the combination of *erf2**Δ* and m*SUR1* improves lactic acid tolerance to some degree. However, the reverse-engineered strain showed significantly lower lactic acid tolerance than the evolved strain. This result suggests that lactic acid tolerance is a complex trait with multiple genetic factors. 

### 3.3. Metabolomic Analysis of the Evolved Strain

To identify metabolomic differences between SR8LDH and BK01, fresh grown cells without lactic acid stress were used for the extraction of intracellular metabolites and analysis by GC-MS. Based on PCA, the intracellular metabolites of SR8LDH and BK01 were clearly separated by PC1, as shown in a score plot (Figure 2A), and the top 10 metabolites that were the most significantly different were classified by PC1, as shown in a loading plot (Figure 2B). Among those significantly different metabolites (Appendix A), galactonic acid, inositol-4-monophosphate, glucose-6-phosphate, glutathione, and maltose were significantly high in BK01, while threonine, citric acid, galactinol, pyrophosphate, and N-acetylglutamate were significantly low in BK01 compared with SR8LDH. These results suggested that SR8LDH and BK01 have global metabolic differences, including sugar metabolism (glucose-6-phosphate and citric acid), lipid biosynthesis (inositol-4-monophosphate), amino acid biosynthesis (threonine and N-acetylglutamate), and stress response (glutathione).

### 3.4. Lactic Acid Production by the Evolved Strain

To determine whether lactic acid-tolerant BK01 could be a better lactic acid producer than SR8LDH, fermentation of a high glucose concentration (200 g/L) was performed. Under such conditions, lactic acid can be accumulated at a level higher than 80 g/L, and 80 g/L lactic acid is a critical concentration at which BK01 shows growth benefits. For this fermentation, no pH-neutralizing agent was used. The rate of glucose consumption was similar between the two strains, although the maximum cell concentration was higher in SR8LDH (Figure 3A). The lower cell concentration of BK01 was not observed during low glucose fermentation (Appendix A). Meanwhile, the maximum lactic acid production of BK01 was 119.1 g/L, which was 17% higher than SR8LDH (102.4 g/L) (Figure 3B). Ethanol was produced at 36.2 g/L and 38.9 g/L for BK01 and SR8LDH, respectively; thus, SR8LDH produced slightly more ethanol than BK01. When the mass balance of fermentation products (lactic acid, ethanol, and CO_2_) from glucose was calculated at 96 h, BK01 showed a higher carbon recovery (93.7%) than SR8LDH (86.0%), which was consistent with higher cell growth in SR8LDH.

When lactic acid accumulated more than 100 g/L during 200 g/L glucose fermentation, the pH of the medium dropped to 3 (Appendix A), lower than its pKa value (3.86). Under such conditions, the lactic acid-tolerant BK01 strain showed higher lactic acid production than SR8LDH, and the highest reported titer (119 g/L lactic acid) was achieved without pH control (Table 2).

The use of neutralizing agents, such as Ca(OH)_2_ and CaCO_3,_ is currently necessary for industrial lactic acid production [23]. During re-acidification for purification of lactic acid, gypsum (CaSO_4_) is produced as a byproduct, and additional processes and costs are required to remove this contaminant [43]. In addition, the huge amount of gypsum produced with the same mass as lactic acid causes serious environmental problems [43,44,45]. Alternatively, ammonia and ammonia hydroxide neutralizing agents are commonly used to avoid formation of gypsum, but they could decrease lactic acid titers because ammonium lactate produced during fermentation can cause osmotic stress to microbial cells [23]. Therefore, the use of a lactic acid-tolerant strain developed by ALE, such as BK01, could be a promising option for minimizing or excluding pH control for lactic acid bioprocesses. 

### 3.5. Cellulosic Lactic Acid Production by the Evolved Strain

To evaluate a potential to use BK01 for cellulosic lactic acid production, SSF was performed using buckwheat husk hydrolysates (Figure 4A). Buckwheat husk hydrolysates were prepared by dilute-acid pretreatment using 1% (*w/v*) H_2_SO_4_ at 121 °C for 30 min. At a high solid loading of the hydrolysates (20%, dry weight basis), SR8LDH and BK01 showed similar fermentation profiles, with pH changes from 6.5 to 4.2 (Figure 4). During 24 h, both glucose (50–55 g/L) and xylose (3–5 g/L) were consumed, and 32–34 g/L lactic acid was produced. On the other hand, the ethanol production of BK01 (17.6 g/L) was significantly higher than that of SR8LDH (14.4 g/L). 

Because the fermentation of 20% buckwheat husk hydrolysates did not accumulate lactic acid to its toxic level (8%), BK01 might have a higher tolerance to fermentation inhibitors in the hydrolysates. Acetic acid is one of the important fermentation inhibitors of cellulosic biomass hydrolysates [48,49], and 20% buckwheat husk hydrolysates contains 5.3 g/L acetic acid. As the fermentation progressed, lactic acid was accumulated, and the pH of the cellulosic biomass hydrolysates dropped from 6.5 to 4.2. At this low pH (lower than the pKa value of acetic acid, 4.75), acetic acid would be the most critical inhibitor of cellulosic fermentation. Therefore, it was speculated that the improved cellulosic ethanol production by BK01 might be associated with improved acetic acid tolerance. Comparison of SR8LDH and BK01 on 20 g/L glucose supplemented with 4 g/L acetic acid confirmed that BK01 has a higher tolerance to acetic acid than SR8LDH (Appendix A). The cell growth, lactic acid production, and ethanol production of BK01 were significantly better than SR8LDH under the acetic acid stress.

Lactic acid production by engineered S. cerevisiae was demonstrated mostly using complex and synthetic medium [50], and cellulosic lactic acid production was reported only in a few recent studies [8,51]. Using spent coffee ground hydrolysates and wheat straw hydrolysates, 11.5 g/L [8] and 10 g/L [51] lactic acid were produced. The increases in media potassium (K^+^) and pH were able to improve cellulosic lactic acid production significantly [51]. Additionally, the overexpression of GRE2 with detoxifying activity toward fermentation inhibitors in cellulosic biomass hydrolysates (furfural and hydroxymethylfurfural) contributed to further improvement in cellulosic lactic acid production. Therefore, the cellulosic lactic acid production by acid-tolerant BK01 is expected to be enhanced by the optimization of cellulosic medium compositions and further strain engineering for other stress tolerance. 

## 4. Conclusions

The lactic acid-producing *S. cerevisiae* strain was subjected to ALE with a high concentration of lactic acid, and a lactic acid-tolerant BK01 strain was successfully isolated. Mutations responsible for the improved lactic acid tolerance of BK01 could not be identified from genome re-sequencing and reverse engineering, which suggests complex molecular mechanisms of the lactic acid tolerance. Meanwhile, BK01 showed distinct metabolomic profiles compared with its parental strain without lactic acid stress, suggesting a global metabolic shift by the evolution. Lactic-acid tolerant BK01 was able to score the highest lactic acid production (119 g/L) from glucose without the use of a pH-neutralizing agent. Interestingly, BK01 showed improved tolerance to acetic acid as well, which contributed to improved fermentation of acetate-rich lignocellulosic biomass hydrolysates. These results suggest that (1) ALE is a useful tool for developing complex traits of industrial interest, (2) lactic acid tolerance significantly improves lactic acid production, and (3) lactic acid tolerance might contribute to tolerance against other organic acids, including acetic acid. 

## Figures and Tables

**Figure 1 jof-07-00928-f001:**
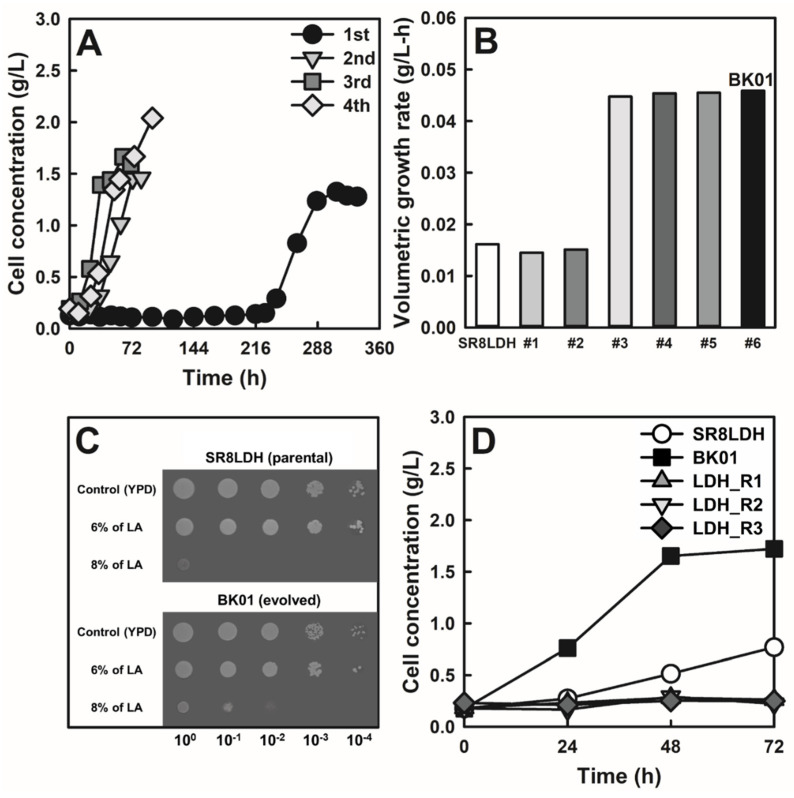
Adaptive laboratory evolution (ALE) of the lactic acid-producing *S. cerevisiae* SR8LDH strain in a high concentration of lactic acid and isolation of the lactic acid-tolerant BK01 strain. (**A**) Cell concentrations were monitored during four consecutive subcultures of SR8LDH in complex medium containing 20 g/L glucose and 8% lactic acid. (**B**) Comparison of volumetric growth rates (g/L-h) of six evolved SR8LDH strains (#1–#6) in 8% lactic acid and the selection of a representative evolved strain, BK01 (#6). (**C**) Spotting assay of SR8LDH and BK01 on complex agar medium containing 20 g/L glucose and 0%, 6%, and 8% lactic acid. (**D**) Growth rate comparisons of SR8LDH, BK01, and the reverse-engineered strains (described in Table 1) in complex medium containing 20 g/L glucose and 8% lactic acid. All experiments were conducted under oxygen-limited conditions (80 rpm), with an initial cell concentration of 0.05 g DCW/L.

**Figure 2 jof-07-00928-f002:**
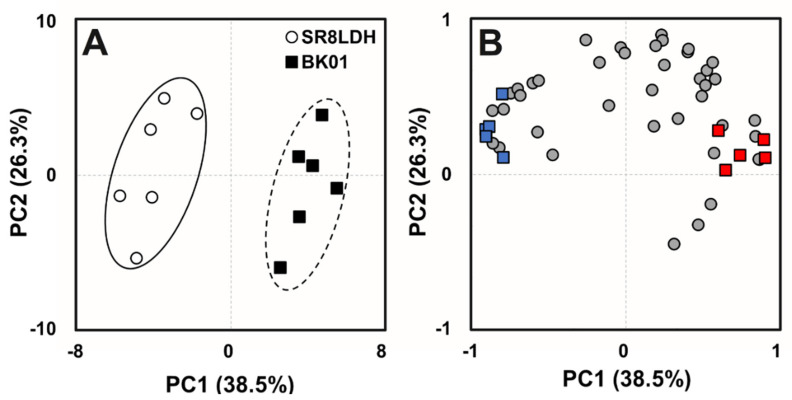
Principle component analysis of significantly different intracellular metabolomic profiles of SR8LDH and BK01 at a mid-exponential growth phase without lactic acid stress on (**A**) a score plot and (**B**) a loading plot. The colored metabolites indicate the five metabolites with the highest (red) and the lowest (blue) fold-change value among significantly different metabolites between SR8LDH and BK01. (Student’s *t*-test, *p* < 0.05), of which details can be found in Appendix A. All cells were grown in complex medium containing 20 g/L glucose, under oxygen-limited conditions (80 rpm), with an initial cell concentration of 0.5 g DCW/L.

**Figure 3 jof-07-00928-f003:**
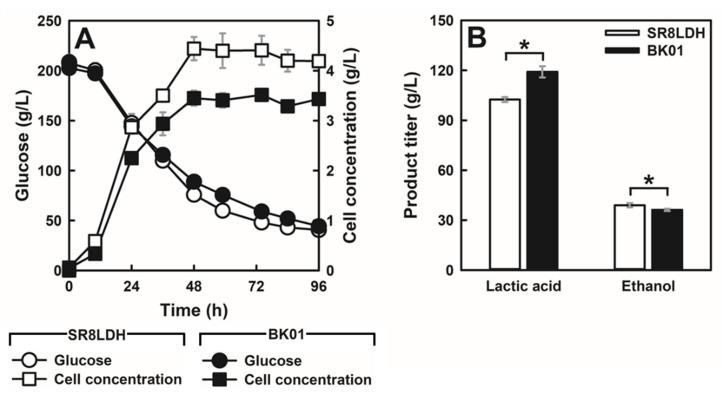
Comparison of lactic acid production by SR8LDH and BK01 in a high concentration of glucose. (**A**) Glucose (circle) and cell concentrations (square). (**B**) Lactic acid and ethanol titers for 96 h of fermentation. The values represent the mean of three independent experiments. The error bars indicate standard deviation. Asterisks denote statistically significant differences (Student’s *t*-test, *p* < 0.05). Fermentations were performed using complex medium containing 200 g/L glucose, under oxygen-limited conditions (80 rpm), with an initial cell concentration of 0.05 g DCW/L.

**Figure 4 jof-07-00928-f004:**
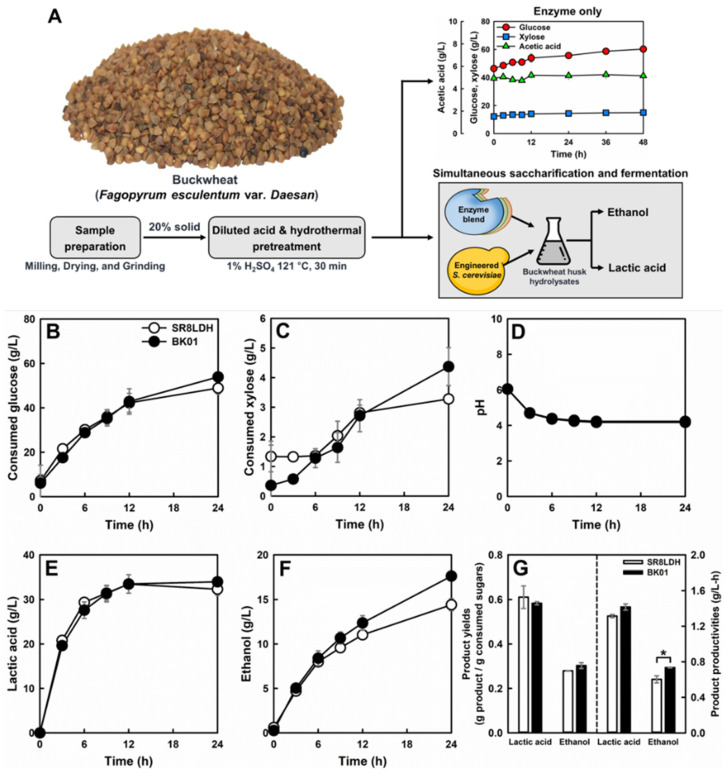
Cellulosic lactic acid production by SR8LDH and BK01 from simultaneous saccharification and fermentation of buckwheat husk hydrolysates (**A**). Consumed glucose (**B**), consumed xylose (**C**), pH (**D**), lactic acid (**E**), ethanol (**F**), and productivities (**G**) were compared. Values represent the mean of three independent experiments. Error bars indicate standard deviation. Asterisks denote statistically significant differences (Student’s *t*-test, *p* < 0.05).

**Table 1 jof-07-00928-t001:** Strains used in this study.

Description	Strain	Genotype	Reference
Background strain	SR8	*S. cerevisiae* D452-2 expressing the xylose oxidoreductase pathway derived from *Pichia stipitis* (*XYL1*, *XYL2*, and *XYL3*), Δ*ald6*, adaptive laboratory evolution on xylose	[29]
Parental strain	SR8LDH	Xylose-metabolizing SR8 expressing the *ldh* gene from *Lactobacillus acidophilus*	[8]
Evolved strain	BK01	SR8LDH strain evolved on complex medium containing 20 g/L glucose and 8% lactic acid	This study
	BK_D1	BK01 Δ*ypt7*	This study
	BK_D2	BK01 *Δyol159c-a*	This study
	BK_D3	BK01 Δ*ypt7*/*Δyol159c-a*	This study
Reverse-engineered strains	LDH_R1	SR8LDH m*YPT7* (533G>A)	This study
	LDH_R2	SR8LDH m*YOL159C-A* (172C>A)	This study
	LDH_R3	SR8LDH m*YPT7*/m*YOL159C-A*	This study
	LDH_D1	SR8LDH Δ*ypt7*	This study
	LDH_D2	SR8LDH *Δyol159c-a*	This study
	LDH_D3	SR8LDH Δ*ypt7*/*Δyol159c-a*	This study

**Table 2 jof-07-00928-t002:** Comparison of engineered *S. cerevisiae* strains producing lactic acid.

Strain	*LDH* Origin	ALE	pH Control	pH	Time (h)	Yield (g/g)	Titer (g/L)	Reference
δpHδLA2-51/dP36	*Lm*	+	20 g/L CaCO_3_	6.4	23	0.55	52	[42]
YPH499/dPdA3-34/DLDH/1-18	*Lm*	−	40 g/L CaCO_3_	ND	216	0.65	60	[16]
JHY5730	*Lm*	+	4 N NaOH	3.5	55	0.83	83	[17]
JHY5330	*Lm*	−	50 g/L CaCO_3_	ND	51	0.80	112	[18]
SP7	*Ps*	−	Continuous	3.5–6.0	49	0.58	117	[15]
T165R	*Bt*	−	Unknown	5.2	48	0.61	122	[14]
SP1130	*Ps*, *Bt*	−	5 N Ca(OH)_2_	4.7	40	0.89	142	[20]
δpHδLA2-51/dP36	*Lm*	+	None	ND	52	0.30	34	[42]
SH6779	*Bt*	−	None	ND	72	ND	48	[46]
YIP-A15G12	*Ec*	−	None	ND	76	0.70	92	[47]
SR8LDH	*La*	−	None	ND	96	0.61	102	This study
BK01	*La*	+	None	ND	96	0.72	119	This study

Lm; *Leuconostoc mesenteroides*, Ps; *Pelodiscus sinensis* japonicas, Bt; *Bos taurus*, Ec; *Escherichia coli*, La; *Lactobacillus acidophilus*. ALE; adaptive laboratory evolution, ND; not determined, +; yes, −; no.

## Data Availability

Publicly available datasets were analyzed in this study. The data can be found here: https://www.ncbi.nlm.nih.gov/bioproject/PRJNA690663.

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
