# Peer review of "l-Lactic Acid Production Using Engineered Saccharomyces cerevisiae with Improved Organic Acid Tolerance"

_jof, 2021, doi:10.3390/jof7110928_

Round 1
Reviewer 1 Report
This paper reports on the isolation of a Saccharomyces cerevisiae strain through adaptive laboratory evolution that has gained high lactic acid tolerance and apparently as a result is able to accumulate a very high titer of lactic acid, higher than any previously reported for unbuffered lactic acid production. This result is interesting for avoidance of the huge gypsum accumulation that is associated with industrial lactic acid production in a Ca(OH)2 buffered medium.
The scientific value of the paper is good but obviously identification of the causative changes would have made the paper scientifically stronger.
The paper has been well prepared and the results are clearly presented and convincing.
There are, however a number of shortcomings that have to be corrected.
- The importance of lactic acid tolerance for lactic acid production has been reviewed previously in the literature but this is not referred to in this paper.
- The English language and grammar need improvement in several places.
e.g.
‘Particularly, BK01 revealed improved tol-23 erance against acetic acid of the hydrolysates, a major fermentation inhibitor of lignocellulosic biomass.’
‘… high glucose fermentation (200 g/L) was performed.’ Fermentation of a high glucose concentration (200 g/L) was performed.’
‘The lower cell concentration of BK01 was not observed during low glucose fermentation’
- It should be mentioned what pathway is used in the SR8LDH strain for xylose utilization.
- It would be useful to include a mass balance so as to have an idea of the yield: 119.1 g/L lactic acid + 36.2 g/L ethanol + ±36.2 g/L CO2 + … = 200 g/L glucose
- ‘During re-acidification for purification of lactic acid, gypsum (CaSO4) is produced as a byproduct, and additional processes and costs are required to remove this contaminant [37].’ The authors should also mention the huge environmental burden caused by the accumulation of mountains of gypsum.
- In Table 2: pH control: replace the - sign with the text: ‘None’. This will make it more clear that there was no pH control. The - sign could also indicate that it was not mentioned in the paper.
- line 315: ‘Buckwheat husk hydrolysates were prepared by dilute acid pretreatment before fermentation.’ This is a bit misleading, since it gives the impression of a slight treatment, while there was also heating, so it was a harsh treatment. Specify better.
- Section 3.5 and Fig. 4: Xylose utilization was not demonstrated or measured. This is a shortcoming compromising the statement in the abstract: ‘Moreover, cellulosic lactic acid production by BK01 was demonstrated using acetate-rich buckwheat husk hydrolysates.’ This gives the impression that both glucose and xylose were utilized.
Reviewer 2 Report
The manuscript ID jof-1424623 described the lactic acid production by engineered S.cerevisae BK01 developed through adaptive laboratory evolution. This production was improved by 17% compared to the parental strain.
Overall description is very reasonable and logical.
In additions, authors evaluated the lactic acid potential production on buckwheat husk hydrolysates without pH control.
Although authors well described results and discussion (except paragraph 3.5, a lot of mistakes), I don’t feel any in-depth scientific insight from this manuscript.
- Line 36-37 natural producers are LAB. Reference [7] describes lactic acid production by engineered cerevisae. Certainly a reference error.
- In general, check and harmonize references (line 401 no page range, line 412 non journal, etc…)
- Paragraphs 2.2 and 2.3, what is the initial pH before fermentation ?
- Discussion should be developed in particular line 263-266 : what are the metabolic pathways impacted ? (more details). It would be interesting to develop the involvement of these 10 (or in part) molecules in metabolic pathways
- To understand the difference in production between the 2 strains, a genomic sequencing and a metabolomic analysis were carried out. An enzymatic physiological study is missing : at least the determination of the inhibition constants (Ki for lactic and acetic acid) and if possible the LDH activity.
- Paragraph 3.5 is not clear
First, figure 4 contains errors : Fig 4B is not described ; in legend and text B is C, C is D etc..
Lines 321-322, the values 15.6 and 9.3 g/L of lactic acid are not in Fig 4. What are these values, at what time ?
Figures 4D and 4F show the product titers : at what time ? The rate (g/l/h) would be more appropriate and the calculation of yields (lactic acid/glucose and lactic acid/biomass) could be interesting as a comparison parameter.
The pH is not control. Could you indicate the pH evolution during fermentation for the 2 strains in figure 4. This information is essential to understand the inhibition problems (pKa, undissociated state of acid…). According to yours results, you may be able to better interpret any acid tolerances
- Table S4 : what is the unit for the indicated averages ?
- Figure S2 : please, indicate the error bars
The manuscript has to be worked to gain some understanding (in particular paragraph 3.5)
Round 2
Reviewer 2 Report
I thank the authors for having answered point by point to my questions and for having improved in the understanding paragraph 3.5
This article can be accepted after adding the two articles cited in their answer but not included in the references
Peetermans, A.; Foulquié-Moreno, M.R.; Thevelein, J.M. Mechanisms underlying lactic acid tolerance
and its influence on lactic acid production in Saccharomyces cerevisiae. Microbial Cell 2021, 8, 111.
Eş, I.; Mousavi Khaneghah, A.; Barba, F.J.; Saraiva, J.A.; Sant'Ana, A.S.; Hashemi, S.M.B. Recent
advancements in lactic acid production - a review. Food Research International 2018, 107, 763-770
Best regards
Author Response
We sincerely appreciate the reviewer for recommending two relevant references. We revised the manuscript by citing them in Page 2, Line 55.